# Impact of Underlying RANS Turbulence Models in Zonal Detached Eddy Simulation: Application to a Compressor Rotor

**Julien Marty * and Cédric Uribe**

ONERA—The French Aerospace Lab, 8 rue des Vertugadins, 92190 Meudon, France; cedric.uribe@outlook.fr
* Correspondence: julien.marty@onera.fr

**Abstract:** The present study focuses on the impact of the underlying RANS turbulence model in the Zonal Detached Eddy Simulation (ZDES) method when used for secondary flow prediction. This is carried out in light of three issues commonly investigated for hybrid RANS/LES methods (detection and protection of attached boundary layer, emergence, and growth of resolved turbulent fluctuations and accurate prediction of separation front due to progressive adverse pressure gradient). The studied configuration is the first rotor of a high pressure compressor. Three different turbulence modelings (Spalart and Allmaras model (SA), Menter model with (SST) and without (BSL) shear stress correction) are assessed as ZDES underlying turbulence model and also as turbulence model of unsteady RANS simulations. Whatever the underlying turbulence model, the ZDES behaves well with respect to the first two issues as the boundary layers appear effectively shielded and the RANS-to-LES switch is close downstream of trailing edges and separation fronts leading to a quick LES treatment of wakes and shear layers. Both tip leakage and corner flows are strongly influenced by the Navier–Stokes resolution approach (unsteady RANS vs. ZDES) but the underlying turbulence modelling (SA vs. SST vs. BSL) impacts mainly the junction flow near the hub for both approaches. ZDES underlying turbulence model choice appear essential since it leads to quite different corner flow separation topologies and so to inversion of the downstream stagnation pressure radial gradient.

**Keywords:** turbulence modeling; zonal detached eddy simulation; compressor; tip flow; corner flow

## 1. Introduction

Within a turbomachinery, by comparison to the flow identified by the primary direction and qualified of primary flow, flows linked to streamwise vorticity generation are called secondary flows [1]. They play a significant role on losses [2] within, performances of and surge margin of compressors. The tip leakage vortex flow can be responsible for rotating instabilities [3], rotating stall [4] or even surge [5], especially if the compressor is tip-critical. The corner separation can also limit the stable operating range and performances of compressor [6]. Thus, the accurate prediction of this kind of flows is essential for a turbomachinery manufacturer. Since these flows usually present unsteady and intermittent turbulent dynamic with highly three-dimensional separation front and topological development, their prediction via RANS method is challenging [7], especially at off-design conditions. On the contrary, Large Eddy Simulation (LES) is a capable candidate [8]. Nonetheless, as its discretization requirements are demanding since 90% of the turbulent fluctuations kinetic energy must be resolved [9] while considered flows Reynolds number can be high, LES cost is still too expensive for design purpose. Thus, LES, especially Wall Resolved LES, are more commonly used for moderate Reynolds number [10]. As a remedy, Hybrid RANS/LES (HRLES) approach is

a trade-off between accuracy and computational costs as, generally, the attached boundary layer is treated by RANS model while the separated boundary layer and wake are simulated with LES [11,12]. This avoids the too expensive cost of LES near wall. Several research groups develop their own HRLES method, such as Spalart [13], Hodara and Smith [14] for global approaches, and Quéméré and Sagaut [15] for zonal techniques which require specific boundary conditions and interfaces between RANS and LES regions. The hybrid RANS/LES method developed at ONERA is the Zonal Detached Eddy Simulation (ZDES) [16]. Several authors applied hybrid RANS/LES methods to investigate secondary flows in turbomachinery and shows their ability to predict them [17–20], as Riéra et al. [21] with the ZDES method. It must be pointed out that Hybrid RANS/LES approaches are confronted to several elementary issues which impact the method accuracy, among them: ⓐ detection of boundary layer edge up to the separation point and forcing a RANS treatment of their whole thickness, ⓑ emergence and growth of resolved turbulent fluctuations across the interface from RANS to LES regions, ⓒ accurate prediction of the separation front due to progressive adverse pressure gradient, and not a geometric singularity. This latter issue ⓒ is particularly important for secondary flows prediction while directly linked to the underlying turbulence modeling. For instance, Mockett [22] investigates the effect of several underlying eddy viscosity turbulence models or an explicit algebraic Reynolds stress model on the flow prediction over the NACA0021 profile at post-stall condition and over the backward facing ramp in rectangular duct (DESider bump) by the DES and DDES methods. For the first configuration, the sensitivity to the underlying RANS turbulence model is quite negligible as the separation is mainly due to geometric singularities which lead to the emergence of shear layers and turbulent fluctuations. On the contrary, for the second configuration, the sensitivity to the underlying RANS turbulence model is significant as the separation depends strongly on the ability of the model to predict the adverse pressure gradient over the ramp and the corner flows. All turbulence models differ from each other and are not in good agreement with experimental data. Thus, it is essential to evaluate the abilities of HRLES methods with challenging configurations and different underlying RANS turbulence models. This remains to be carried out with ZDES method. Although this approach is historically based on Spalart–Allmaras RANS model, recent developments allow the use of other RANS turbulence models based on the Boussinesq hypothesis [23]. Moreover, in studies dealing with (D)DES methods applied to turbomachinery flows and mentioned earlier, the authors focused more on validation and comprehension of flow physics than on issues ⓐ to ⓒ. In light of these issues, the present study aims at evaluating and confronting several ZDES [16] and unsteady RANS simulations of a turbomachinery configuration with different underlying turbulence models: Spalart–Allmaras (named SA model hereafter, Spalart and Allmaras [24]), Menter without Shear Stress Transport correction (named BSL model hereafter for BaSeLine version, Menter [25]) and with SST correction (named SST model hereafter, Menter [25]). The configuration chosen is the first rotor of a high pressure compressor which purposely exhibits tip and corner secondary flows. Moreover, it has been the subject of an experimental measurement test campaign on which Riéra et al. [21] assessed the ZDES method based on SA model. This latter work will be used and compared to ZDES-BSL, ZDES-SST, and unsteady RANS-SA, RANS-BSL, and RANS-SST computations. Issues ⓐ to ⓒ will be scrutinized and computations will be compared to experimental data. Obviously, in order to highlight the dependency or not of secondary flow prediction to the underlying turbulence models with the ZDES approach, both tip and corner flows will be analyzed. After explaining in detail the experimental facility, the numerical setup and the ZDES approach, the study focuses on flow near walls such as on tip leakage and corner flows.

## 2. Experimental Facility

The simulated rotor is the first one of the 3.5 stage axial research compressor CREATE located at the Laboratory of Fluid Mechanics and Acoustics (LMFA) in Lyon, France and designed by SAFRAN AIRCRAFT ENGINES [5,26]. This compressor is representative of the median stages of modern high pressure compressors and comprises three and a half stages as shown in Figure 1. The number of

stages was chosen to have a magnitude of the secondary flow effects similar to a real compressor, and to be within the rig torque power limitation. The circumferential periodicity of the whole machine (obviously $2\pi$ in the general case with prime blade numbers to avoid generating resonant waves) has been reduced to $\frac{2\pi}{16}$ on the compressor CREATE, choosing the number of blades of each rotor and stator [inlet guide vane (IGV) included] as a multiple of 16. The blade number for each row is detailed in Table 1. Thus, the analysis of rotor–stator interaction does not require measurements over a full annulus sector as a sector of only 22.5° ($\frac{2\pi}{16}$) should contain all the useful spatial information, at least for stabilized operating point. The Reynolds number based on the rotor chord at mid-span is close to 800,000. The tip Mach number is equal to 0.92 and the rotational speed is 11,543 rpm. The casing diameter is 0.52 m.

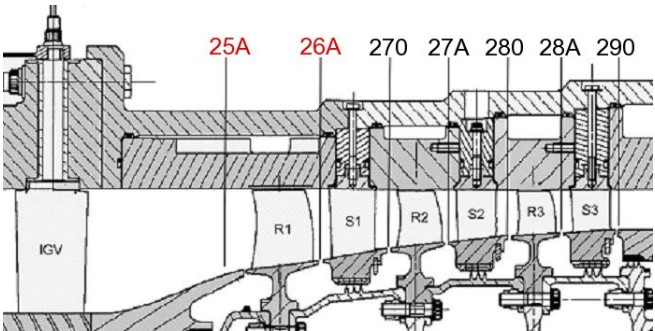

**Figure 1.** CREATE compressor meridian view.

**Table 1.** Blade number for the rows.

| Row | IGV | R1 | S1 | R2 | S2 | R3 | S3 |
|---|---|---|---|---|---|---|---|
| Blade number | 32 | 64 | 96 | 80 | 112 | 80 | 128 |

The meridian view shown in Figure 1 highlights measurement planes between rotor and stator. These measurements are carried out with both pneumatic four-hole probes and unsteady stagnation pressure probes allowing the calculation of overall performances (pressure ratio, efficiency), stall line position included, through azimuthal and radial average, and radial profiles via azimuthal average. At several operating points (near maximum efficiency, near choke and near surge), conventional pressure, temperature, and angle probe traverses were performed downstream of each blade and vane. The four-holes probes had dual pressure and temperature sensors, together with two extra static pressure sensors (mounted on each side of the probe) providing the flow angle when balanced. Thus, the Mach number and also the three velocity components can be computed. The unsteady stagnation pressure probes developed by VKI have a very high bandwidth (250 kHz) and their diameter is 2.5 mm (2.65% of the spatial periodicity of the whole compressor at casing) [27].

## 3. Numerical Test Bench

In the present study, only the first rotor of the compressor (R1) is investigated at the nominal rotational speed and the nominal operating point. The inlet boundary is set to section 25A of the experimental test rig as measurements are available for different operating points at this section. In addition, the possible comparisons with experimental data at section 25A make the numerical setup more reliable for the evaluation of the method. The outlet boundary is defined at two axial chords downstream the R1 in order to have the experimental section 26A within the computational domain and to have a quite axisymmetric outflow as boundary condition is based on this hypothesis (radial equilibrium). The main simplifications concern the two axial gaps at the hub which are not included in the computational domain. The stage matching in a multi-stage compressor is influenced by leakage (due to tip clearances at the casing or axial gaps at the hub) and corner flows. In the present case,

the hub cavity is not taken into account, which can explain locally some discrepancies near the hub. However, as mentioned hereinafter, inlet boundary conditions are set so that the single rotor operating point matches as much as possible the experimental one. Thus, the rotor is as much as possible on target with test-rig data.

The computational grid is obtained with a multi block approach using an O4H topology. The rotor tip clearance is meshed with an additional O-H block. Two rotor blade passages are simulated in order to fit the spatial periodicity ($\frac{2\pi}{32}$) of the IGV (Inlet Guide Vane). The whole grid comprises 88 million points. The normalized wall cell dimension normal to the wall fulfills $\Delta y^+$ of the order of 1 in every zone. In the vicinity of blade walls, $\Delta x^+$ and $\Delta z^+$ are respectively of the order of 200–300 and 100. Downstream of the blades, the mesh is progressively coarsened to 1700 for $\Delta x^+$ and 150 for $\Delta z^+$ to avoid numerical reflections. The normalized wall cell dimensions $\Delta x^+$ and $\Delta z^+$ are detailed in Riéra et al. [21].

The IGV effects are reproduced with specific inlet boundary conditions based on 2D cartography resulting from a previous RANS IGV-R1 computation as measurements are not sufficient to have the whole cartography since experimental data are only available from 6 to 95% of passage relative height in spanwise (i.e., radial) direction. It should be noticed that this inlet condition is not combined with a synthetic eddy method to generate incoming turbulent fluctuations so that boundary conditions are identical between RANS and ZDES computations. Thus, there are no turbulent flow fluctuations at the inlet, in the present study. It consists of a rotating cartography of stagnation pressure, stagnation enthalpy, flow direction and turbulent variables, based on a Fourier decomposition with 60 harmonics of the two-dimensional map of the flow. Similarly to the phase-lag approach, for each radius, the decomposition in Fourier series is performed along azimuthal axis in order to build the inflow condition at each time step. The 2D CFD and experimental cartographies are shown in Figure 2 while radial profiles obtained after an azimuthal average at inlet plane 25A are plotted in Figure 3. Vortices and wake are well reproduced, especially in terms of azimuthal location. In the 25A plane, the radial profile of axial momentum is well reproduced. Nonetheless, the stagnation pressure profile highlights an overestimation of the deficit in tip and root IGV vortices. This is one of the limitations of the numerical test bench representativeness discussed by Riéra [28]. For the outlet boundary condition, the static pressure is specified at the hub and is forced to follow a simplified radial equilibrium law elsewhere. This hub static pressure is adjusted in order that the radial profile of axial momentum at the inlet matches the experimental one near the casing. A classic rotation periodicity condition is set at the azimuthal boundaries and a no-slip adiabatic wall condition is applied at all wall surfaces. More details on boundary conditions are given in Riéra et al. [21].

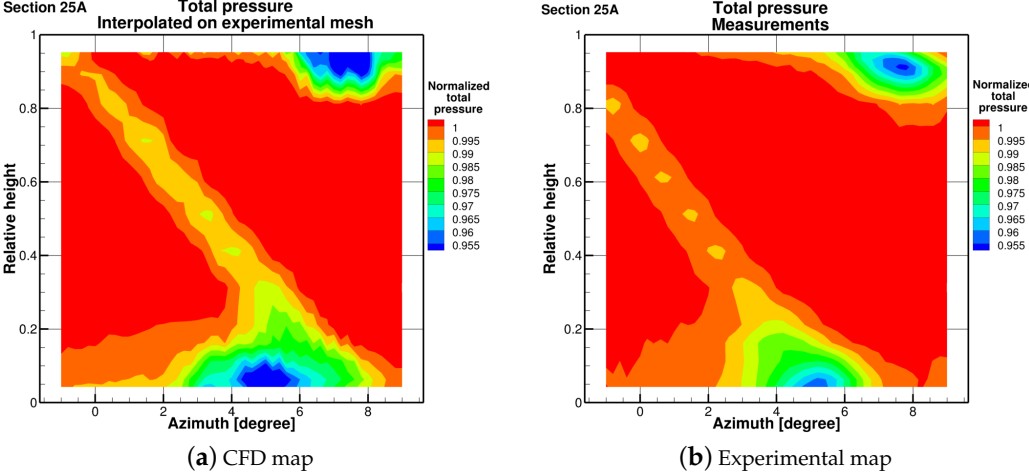

(**a**) CFD map　　　　　　　　　　　　　　　　　(**b**) Experimental map

**Figure 2.** 2D cartographies in plane 25A.

The numerical simulations have been performed using the *elsA* software [29], developed at ONERA and co-owned by AIRBUS, SAFRAN, and ONERA. This compressible flow solver is based on a cell centered finite volume technique and structured multiblock meshes. The spatial discretization scheme for the inviscid fluxes [30] is based on the second order accurate Advection Upstream Splitting Method for low Mach numbers (AUSM+P) associated with third order flux reconstruction with MUSCL technique. The viscous fluxes are computed with a second-order centered scheme. For efficiency, an implicit time integration is employed to deal with the very small grid size encountered near the wall. The time discretization scheme is the second order accurate Gear scheme. At each time step, an approximate Newton method based on the LU factorization solves the nonlinear problem. The time step is set to $1.6 \cdot 10^{-7}$ s, which leads to a Courant–Friedrich–Levy number lower than 1 except for the boundary layers. It corresponds to 1000 time steps per IGV passing period. In the present study, all computations are performed with a unique numerical setup: mesh, scheme, and boundary conditions.

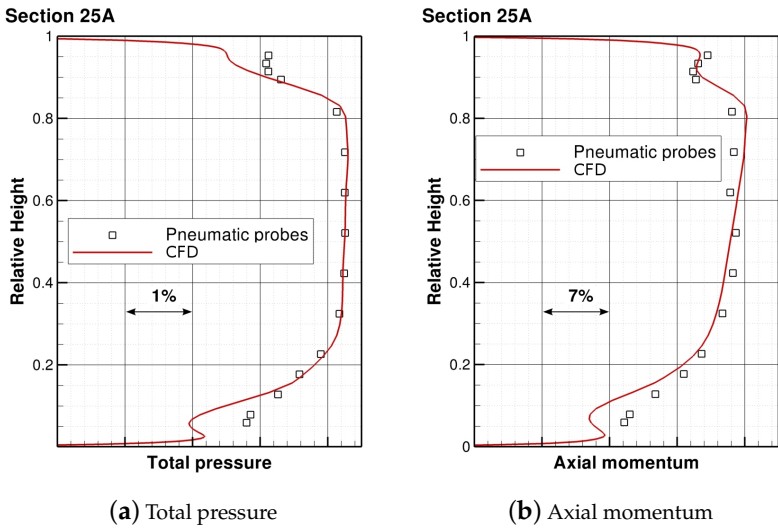

(**a**) Total pressure    (**b**) Axial momentum

**Figure 3.** Radial profiles at inlet plane 25A.

## 4. Zonal Detached Eddy Simulation

The hybrid RANS/LES method called Zonal Detached Eddy Simulation and developed at ONERA [16] is a global approach inherited from Detached Eddy Simulation [13] and Delayed Detached Eddy Simulation [31] approaches. Like the latters, ZDES was originally formulated from a RANS equation system closed by the turbulence model of Spalart and Allmaras [24]. Whatever the underlying turbulence model, the idea is the same as for DES and DDES: thanks to an adequate length scale substitution in the terms of the turbulence model transport equations, the destruction of modeled turbulence can be appropriately increased in order to recover an LES-like behavior since the level of modeled turbulence is in the order of the one produced by a classical LES SubGrid-Scale model (e.g., Smagorinsky [32] model). The switch from RANS behavior to LES one is only done in some regions of the computational domain in order to reduce the computational cost compared to a LES approach (by reducing the spatial and temporal discretization requirements while defining the computational mesh and numerical parameters). Typically, the length scale substitution is based on local flow and mesh properties (e.g., velocity gradient, vorticity, subgrid scale) in such a way that boundary layers are automatically solved using the unmodified original RANS equations while a LES behavior is recover as soon as possible far from the wall and in separated boundary layers. The ZDES approach differs from original DES and DDES approaches by the quickness of this recovering thanks to a particular subgrid scale evoked further. Also, the ZDES approach can be applied using (or not) its zonalization capacity: this feature allows the user to a priori specify the formulation (or mode) the

approach must adopt for any user-defined zones of the computational domain. Those modes differ by their fidelity level and/or by their suitability to the local flow topology. For lowering discretization requirements—and so fidelity level—ZDES can be forced to adopt a RANS behavior in an entire zone: this is the mode 0. For automatically detecting and treating attached boundary layers in their whole thickness with a RANS behavior, and separated flows with a LES behavior, as in DES or DDES, mode 1 or 2 is respectively applied. Let's note that mode 2 is the default mode of ZDES and, as an evolution of mode 1, is preferred to the latter. Finally ZDES can be used locally as a Wall Modeled Large Eddy Simulation approach or a Wall Resolved Large Eddy Simulation approach thanks to mode 3, which is not used in this study. In consequence, each ZDES mode has a domain of predilection in respect to the topology of the computed flow, as shown in Figure 4. Since its formulation is similar to DES, mode 1 has to be used for massive separation prediction whose onset is a priori known, generally provoked by a geometric singularity as in backward facing step configuration. Thanks to an effective detection of boundary layers up to their separation front (similar to DDES), mode 2 can replace mode 1 and also be used for massive separation prediction whose onset is a priori unknown and generally induced by progressive adverse pressure gradient as on a curved surface. Finally mode 3 is dedicated to shallow separation highly sensitive to the internal dynamics of the incoming boundary layer which need to be partially resolved, at a cost out of the scope of this study. A massive separation is here defined by a height significantly higher than the boundary layer thickness at separation point while a shallow separation height is less than or equal to the boundary layer thickness. The zonalization principle (mode 0, 1, 2 or 3) can be illustrated by the work of Deck and Laraufie [33] on a high-lift configuration. They used mode 1 for slat and flap coves as separation is driven by the geometric singularity, mode 2 around the last element as separation is unknown a priori and mode 3 in the vicinity of the middle element trailing edge as small separation depends on the incoming boundary layer. Mode 0 is applied elsewhere.

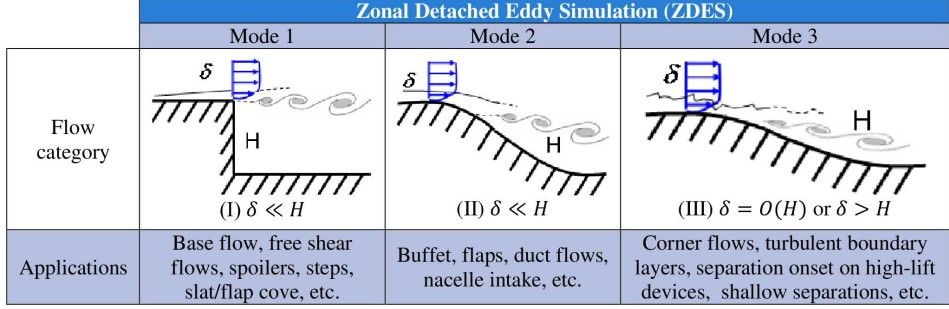

| | Zonal Detached Eddy Simulation (ZDES) | | |
|---|---|---|---|
| | Mode 1 | Mode 2 | Mode 3 |
| Flow category | (I) $\delta \ll H$ | (II) $\delta \ll H$ | (III) $\delta = O(H)$ or $\delta > H$ |
| Applications | Base flow, free shear flows, spoilers, steps, slat/flap cove, etc. | Buffet, flaps, duct flows, nacelle intake, etc. | Corner flows, turbulent boundary layers, separation onset on high-lift devices, shallow separations, etc. |

**Figure 4.** Classification of typical flow problems. I: separation fixed by the geometry, II: separation induced by a pressure gradient on a curved surface, III: separation strongly influenced by the dynamics of the incoming boundary layer (adapted from Deck [16]).

Three ZDES computations will be considered in this study. They differs by their underlying turbulence modeling. A first one, named ZDES-SA hereafter, corresponds to the original formulation of ZDES based on the Spalart–Allmaras turbulence model and detailed in Deck [16]. Two others, named ZDES-SST and ZDES-BSL hereafter, correspond to a formulation of ZDES based on the $k - \omega$ turbulence model of Menter [25], respectively with (ZDES-SST) and without (ZDES-BSL) the Shear Stress Transport correction proposed by Menter [25] (BSL for BaSeLine version of the model). The formulation of ZDES-SST relies on the proposition of Strelets [34] for DES approach, and is detailed by Uribe et al. [23]. It must be pointed out that in this previous work the ZDES-SA and ZDES-SST approaches were assessed on three generic flow test cases (a mixing layer, a backward facing step and a low Reynolds circular cylinder flow) without revealing any significant discrepancy between their predictions. This was expected and explained by the minor role of the underlying turbulence model regarding the test case physics (e.g., absence of turbulent boundary layer separation with unknown

onset location). Also, as the Shear Stress Transport correction aims to improve the boundary layer development prediction in case of adverse pressure gradient [25], it has been chosen to disable it for creating a third formulation which will enriches the following assessment of ZDES underlying turbulence modeling impact. Hence comes the ZDES-BSL approach. Let's note that, as other limiters, the SST correction is only applied when the ZDES-SST works in a RANS behavior (despite an equation misprint in [23] indicating the opposite). Neither vorticity based production terms nor vorticity based SST correction are used.

　　The main terms of the three selected ZDES approaches are summarized in Table 2, for modes 1 and 2. In this table, the mode number corresponds to the one defined in Figure 4 and all terms referred to the original transport equations of RANS model. The building of a ZDES approach is briefly recalled here: the RANS equation system closed by the chosen turbulence model is the starting point. The RANS length scale ($L_{RANS}$) of the turbulence model is expressed (if not obvious as the wall distance $d_w$ for ZDES-SA) thanks to turbulent variables (as the $\frac{\sqrt{k}}{\beta^\star \omega}$ ratio for ZDES-SST and ZDES-BSL) and its appearances are identified in the equations. Some of them are replaced by an hybrid length scale ($L_{DES}$), classically in destruction term of the first turbulent variable, as here, but several choices have been proposed [23]. In Table 2 $f_w$ and $\tilde{S}$ are SA model terms relative to wall dumping functions. Whatever the mode, when $L_{DES}$ equals $L_{RANS}$ a RANS behavior (or at least the original equation system) is recovered, whereas when $L_{DES}$ equals the LES length scale $L_{LES}$, a LES behavior is expected since, as explained previously, (i) the level of modeled turbulence is then in the order of a LES SubGrid-Scale Smagorinsky [32] model ones and (ii) the birth and development of resolved turbulent fluctuations are then possible (weaker damping) and supposed to appears quickly (e.g., transitional inertial instabilities from existing velocity gradients). Thus, respect of discretization requirements is essential to promote a correct LES behavior. The expression of $L_{DES}$ depends of the selected mode and aims at switching as quickly as possible, but continuously, between $L_{RANS}$ and $L_{LES}$ values. Also, this switching has to be done automatically at the right locations, since attached boundary layers have to be encompassed in a so called RANS region where $L_{DES}$ equals $L_{RANS}$, whereas secondary flows as tip vortex and separation have to be encompassed in a so called LES region where $L_{DES}$ equals $L_{LES}$. That why, as remedy to the mesh dependency of the mode 1 switching [31], the $L_{DES}$ term for mode 2 relies on a so called protection function (generic name $f_p$) identifying the frontiers of attached boundary layers. Since $f_p$ is based on turbulent variables, a particular underlying turbulence model is generally associated with a particular protection function, as $f_d(\tilde{v})$ for ZDES-SA or $1 - F_F$ for ZDES-BSL and ZDES-SST here. Finally, it has to be noticed that the LES length scale $L_{LES}$ is expressed as the product of a constant ($C_{DES}$) by a subgrid (length) scale ($\Delta^I_{DES}$ or $\Delta^{II}_{DES}$ depending of the mode). Hence the subgrid modeled turbulence produced by ZDES approaches in LES regions is proportional to the local grid cell dimensions, as for DES, DDES and classical LES subgrid-scale model. However, on this point, ZDES differs from the latters by the use of a vorticity based subgrid scale $\Delta_\omega$ [16,35] allowing a quicker switching from RANS to LES behavior. For all modes, the $C_{DES}$ value comes from calibration against decaying homogeneous isotropic turbulence measurements [36].

**Table 2.** Differences between ZDES approaches depending on the underlying turbulence model and the selected mode (mode numbers are relative to those defined in Figure 4). $d_w$ is wall distance and all terms refer to original RANS formulation (e.g., $D_k$ or $D_{\tilde{v}}$).

| | ZDES SA | ZDES BSL and ZDES SST |
|---|---|---|
| $L_{\text{RANS}}$ | $d_w$ | $\frac{\sqrt{k}}{\beta^\star \omega}$ |
| Substitution of $L_{\text{RANS}}$ by $L_{\text{DES}}$ in terms | $\begin{cases} D_{\tilde{v}} \\ f_w \\ \tilde{S} \end{cases}$ | $D_k$ |
| $C_{\text{DES}}$ | 0.65 | $\begin{cases} (1 - F_1)\, C_{\text{DES}}^{k-\varepsilon} + F_1 C_{\text{DES}}^{k-\omega} \\ C_{\text{DES}}^{k-\varepsilon} = 0.61 \\ C_{\text{DES}}^{k-\omega} = 0.78 \end{cases}$ |
| | **Mode 1** | |
| $L_{\text{DES}}$ | $\min\left(L_{\text{RANS}}, L_{\text{LES}}\right)$ | |
| $L_{\text{LES}}$ | $C_{\text{DES}}\Delta_{\text{DES}}^{I}$ | |
| $\Delta_{\text{DES}}^{I}$ | $\Delta_{\text{vol}}$ or $\Delta_\omega$ (user choice) | |
| | **Mode 2** | |
| $L_{\text{DES}}$ | $L_{\text{RANS}} - f_p \max\left(0, L_{\text{RANS}} - L_{\text{LES}}\right)$ | |
| $L_{\text{LES}}$ | $C_{\text{DES}}\Delta_{\text{DES}}^{II}$ | |
| $\Delta_{\text{DES}}^{II}$ | $\begin{cases} \Delta_{\text{max}} & \text{if } f_p \leq f_{p_0} \\ \Delta_{\text{vol}} \text{ or } \Delta_\omega & \text{if } f_p > f_{p_0} \end{cases}$ | |
| $f_p$ | $f_d\left(\tilde{v}\right)$ | $\begin{cases} 1 - F_F = 1 - tanh\left(\xi^4\right) \\ \xi = \max\left(\frac{\sqrt{k}}{\beta^\star \omega d_w}; \frac{500v}{\omega d_w^2}\right) \end{cases}$ |
| $f_{p_0}$ | 0.8 | 0.8 |

## 5. Rotor Performances

Figure 5 shows the radial profile of absolute deviation angle $\alpha$ and stagnation pressure $p_{i_a}$ in the experimental plane 26A located downstream the first rotor. Five numerical simulation predictions are confronted to experimental data obtained with pneumatic probes and unsteady pressure probes [26]. Two unsteady RANS simulations are performed for comparison: RANS-SA and RANS-SST while three ZDES simulations are performed with SA, BSL, and SST underlying turbulence models. As all simulations are unsteady, these radial profiles are obtained after a time-average over twenty IGV passing periods and then an azimuthal average. As previously mentioned, there is no hub cavity in numerical simulations.

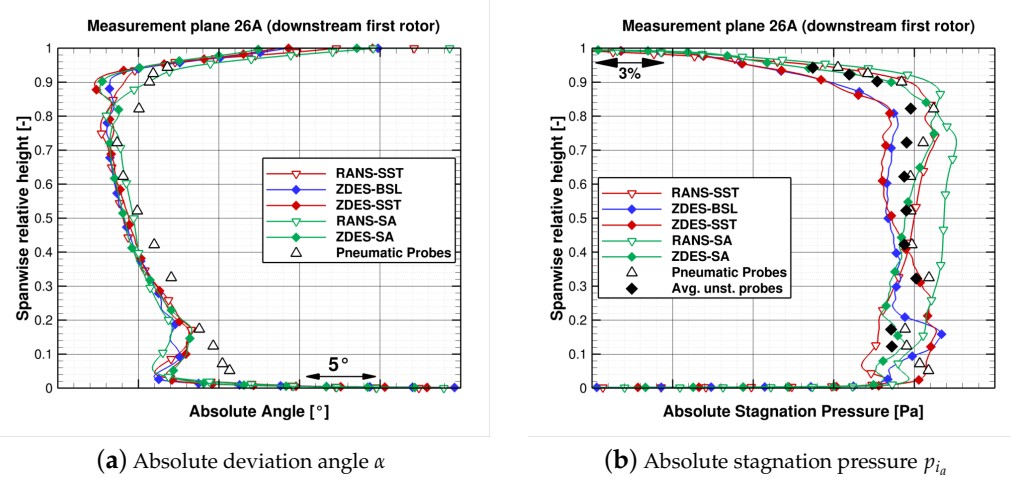

(**a**) Absolute deviation angle $\alpha$  (**b**) Absolute stagnation pressure $p_{i_a}$

**Figure 5.** Radial profiles in plane 26A.

The comparison of numerical results to experimental data obtained using pneumatic probes (4-hole probes) show a good agreement between them. Levels and gradients are well captured by CFD. For absolute deviation angle, the most significant discrepancies are observed near the casing and the hub. Close to the hub, this is due to the absence of leakage flow caused by axial gap between fixed and rotating parts of the hub. This gap is responsible for a flow injection characterized by a strong azimuthal component which interacts with the main flow and modifies locally the deviation angle. The discrepancy is also due to the underestimation of axial momentum in plane 25A (Figure 3b) which is responsible for a mismatch in terms of flow deviation close to the hub. Near the casing, the discrepancies are due to the different prediction of tip leakage flow, especially the tip leakage vortex and its behavior across the weak shock, and its role in the double leakage phenomenon, as shown later. All ZDES predict an underdeviation of a few degrees.

The agreement of absolute stagnation pressure profile with experimental data is not as good as the absolute deviation angle. The phenomena of tip leakage flow, double leakage flow, thickening and separation of blade boundary layer, investigated in the next sections, influence more the stagnation pressure than the deviation angle. Nevertheless, discrepancies remain quite small (less than 1.5%) and close to hub (spanwise relative height, h/H, below 10%), the main causes of discrepancy are the underestimation of stagnation pressure by 1% in plane 25A (Figure 3a) and the absence of hub cavity. The thickening and then separation predicted by ZDES-BSL and ZDES-SST are responsible for stagnation pressure losses above 30% h/H. Nonetheless, despite the loss overestimation, these simulations are able to well capture the radial gradient of stagnation pressure, especially the negative gradient between 30% h/H and 50% h/H. Near the hub, the stronger and more azimuthally spread mixing of separated boundary layer predicted by RANS-SST and ZDES-SA—shown further with Figure 16—leads to an overestimation of losses. Beside these disparities, the analysis of radial profiles shows an overall good agreement of all numerical simulations with experimental data which validates their use for the following investigations. In the next sections, discrepancies between ZDES simulations are investigated in more detail.

Figure 6 shows the rise of absolute stagnation pressure and temperature across the first rotor. Experimental data are based on pneumatic probes as unsteady probe data are not measured in 25A plane. The comparison to experimental data highlights the good agreement of numerical results, especially the ZDES-SA ones which leads to the best matching all along the span. The discrepancies between all other numerical results and experimental data are smaller than 3%. The main differences against the experimental data are observed near the hub for RANS simulations based on both SA and SST model and for ZDES-BSL and ZDES-SST simulations. As explained in the next sections, this is due to the prediction of the flow at the blade–hub junction and of the corner separation. It should be

noticed that the prediction of corner flow is also influenced by the absence of hub cavity in simulations. The analysis of RANS results shows that the corner separation is too massive with SST model. On the contrary, the overestimation of the stagnation pressure ratio near the hub for ZDES simulations based on $k - \omega$ SST turbulence model is caused by a smaller separation at the corner. Concerning the ZDES-BSL simulation, the strong reduction of stagnation pressure rise near the hub (spanwise relative height smaller than 10%) is due to the small but intense separation. Both ZDES-BSL and ZDES-SST underestimate the stagnation pressure rise from mid-span to the casing. This is due to the boundary layer development over the rotor blade and of the corner separation which is less spread in azimuthal direction but more spread spanwise. With respect to the stagnation temperature ratio, numerical results match experimental data well as the error is smaller than 1% and the radial gradient is well captured. Main discrepancies are observed near the casing where numerical results obtained with RANS approach is closer to experimental data levels but fail to predict correctly the radial gradient i.e., the decrease of stagnation temperature ratio near the casing. Only ZDES simulations, especially ZDES-SA, is able to capture this gradient. This is due to the different behavior of approach in the casing boundary layer in which the turbulence is modeled by RANS (RANS-SA, RANS-SST) or solved by the LES part of the ZDES since the tip leakage flow causes the switch from RANS to LES behavior in the vicinity of the casing for the whole circumference. It should be noticed the stagnation temperature rise profile can be also influenced by the thermal boundary condition. The assumption of adiabatic wall at the casing is possibly not correct.

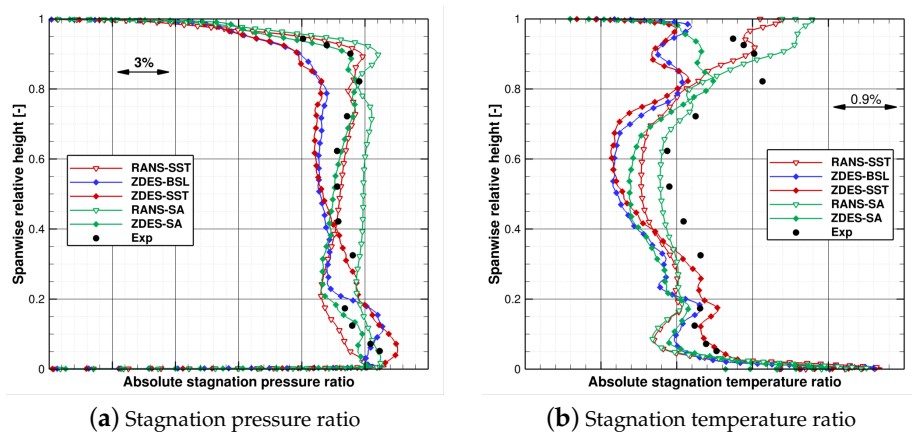

(**a**) Stagnation pressure ratio　　　　　　　　(**b**) Stagnation temperature ratio

**Figure 6.** Stagnation pressure and temperature ratios between planes 25A (upstream) and 26A (downstream).

Figure 7 depicts the profiles of the normalized axial and circumferential velocities and their respective standard deviations at the intersection of 26A plane with 90% spanwise relative height isosurface. Experimental data are obtained through LDA measurements and phase average. Thus, the influence of the IGV is still visible and explains the non periodicity between the two blade channels, especially the peak at 2.75° in azimuthal velocity profile. It should be noticed that, due to the CPU cost of phase average in simulations, all numerical simulations are time-averaged, subsequently removing the IGV footprint in CFD curves. It is reminded that the simulations are performed over two blade channels and the time average is applied on the whole computational domain. This explains small discrepancies in CFD profiles between the two blade channels. All simulations predict correctly both axial and circumferential components of velocity measured by LDA. The RANS simulations overestimate the circumferential velocity in the wake characterized by the peak (and the axial velocity deficit) while the ZDES simulation based on SST turbulence model underestimates the peak level. Outside the wake region, all velocity profiles are well predicted. Whatever the component, the location of the wake is well predicted by simulations. With respect to the standard deviations, as already

shown by Riéra et al. [21], only ZDES simulations are able to capture the velocity fluctuations due to the wake (main peak) and to the tip leakage vortex (second peak). All RANS simulations are able to capture neither the rise of fluctuations in wakes nor the wake thickness in standard deviations profiles. Circumferential gradients and levels are not captured by RANS simulations. This is expected as RANS approach models the whole turbulence spectrum and is not well suited to predict fluctuations while the ZDES approach solves partially the turbulence spectrum once the approach behaves as an LES (e.g., in the tip region here). The overestimation of standard deviations by all ZDES simulations can be explained by—on CFD side—the reduced computational domain and the absence of downstream stator responsible for potential effects but also by—on experimental side—the spatial average caused by the ellipsoid measurement volume of which the semi-major axis is equal to 2% of relative height and the azimuthal extent to 0.7% of the pitch. Thus, experimental data can be considered as radially averaged leading to an underestimation of the fluctuations and to a shift in terms of radial and/or azimuthal locations of vortices.

Finally, despite the absence of the hub gap and the subsequent leakage flow and of the downstream stator, numerical simulations are close to experimental data, validating the use of those simulations to understand the influence of the underlying RANS turbulence model in the ZDES approach.

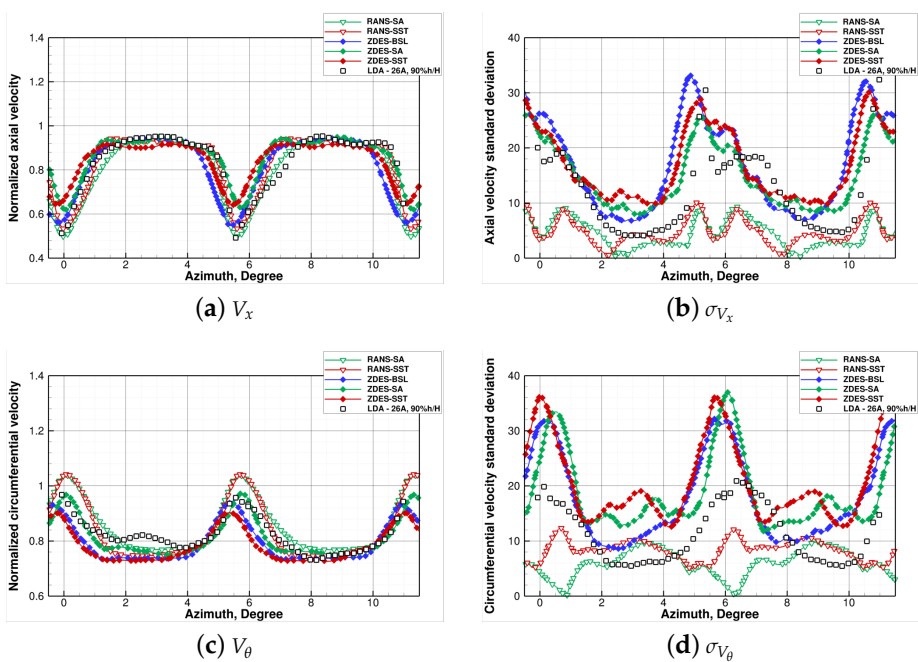

**Figure 7.** Axial ($V_x$) and circumferential ($V_\theta$) components of velocity $V$ and their respective standard deviations $\sigma$.

## 6. Flow around Rotor Blades

The first problematic mentioned in introduction Ⓐ concerns the protection of boundary layers which enables the latter to be treated with RANS equation system in their whole thickness while flow far from the walls and separated flows must be solved with LES behavior. Note that mode 2 is used in the tip gap (O-H topology), as shown by the ZDES mode distribution depicted in Figure 8. In order to visualize the kind of used equations (RANS or LES) and the grey zones with intermediate treatment, the $\lambda$ sensor is defined as follows:

$$\lambda = \frac{L_{\text{DES}} - L_{\text{RANS}}}{L_{\text{LES}} - L_{\text{RANS}}} \tag{1}$$

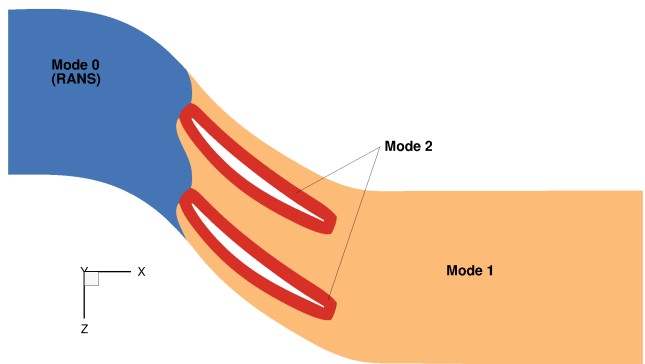

**Figure 8.** ZDES mode distribution in blade-to-blade view. The mode 2 is used in the tip gap (O-H topology).

This sensor is equal to zero in RANS regions, one in LES regions, and ranges from zero to one in grey zones. Figure 9 depicts the contours of $\lambda$ sensor at mid-span, at the instant $t = \frac{T}{2}$ and isolines of entropy variation colored by ZDES mode (0, 1 or 2), for the ZDES simulations based on Spalart–Allmaras model (ZDES-SA), BSL and SST $k - \omega$ Menter model (ZDES-BSL and ZDES-SST). The entropy variation highlights the IGV wakes and growth of boundary layers over the rotor blades. The instant $t = \frac{T}{2}$ is not arbitrarily chosen. This is the instant where the IGV wakes impinge the blade in foreground while the blade visible in background is far from this interaction. This instant is relative to two extrema of the IGV wake passing period. Thanks to the entropy variation isolines, the convection of IGV wakes is clearly observed. These wakes are treated by LES once they enter in regions relative to ZDES modes 1 and 2. The mode choice has no influence on the IGV wakes. Due to the formulation of $L_{\text{RANS}}$ based on $k$ and $\omega$, some RANS islands are visible in ZDES-BSL and ZDES-SST predictions. These RANS islands seem to not have significant effect on both boundary layer protection and LES region appearance as they only concern the flow field far from both IGV and rotor wakes. The development of the near-wall RANS regions and the positioning of entropy variation isolines along chord suggest a monotonic growth of the RANS boundary layers up to the separation points, as expected for effectively protected boundary layers. The ZDES-SA prediction differs from the two others as the boundary layer is protected up to the trailing edge. For the ZDES-BSL and ZDES-SST simulations, the RANS region is less spread close to the trailing edge. As shown later, this is due to a significant discrepancy in corner separation which is not observed with ZDES-SA at mid-span while both ZDES-BSL and ZDES-SST show a boundary layer separation. This leads to a thicker apparent boundary layer for these two simulations as shown by entropy variation isolines. On pressure side, the boundary layer seems to be accurately protected, without any evidence of unwanted grey zone. It should be noticed that, directly downstream from the trailing edge, both IGV and rotor wakes are treated with LES equation system. This allows an early apparition and growth of resolved turbulent content.

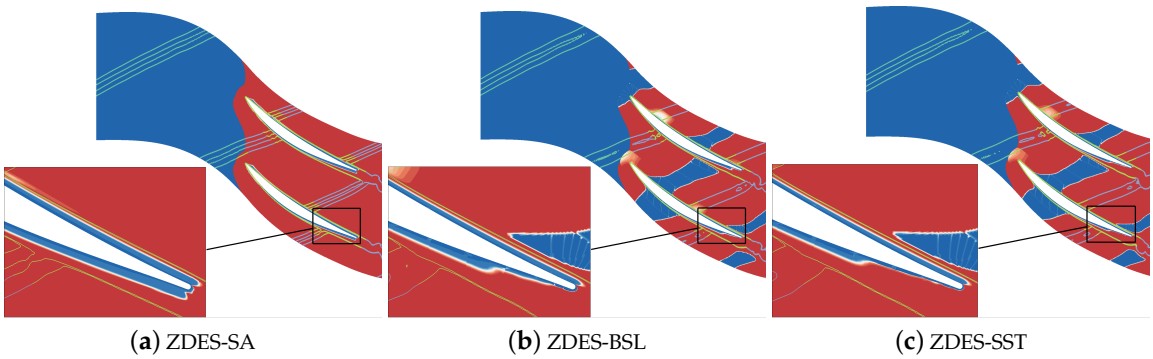

(**a**) ZDES-SA                                    (**b**) ZDES-BSL                                    (**c**) ZDES-SST

**Figure 9.** Contours of $\lambda$ sensor at mid-span at the instant $t = T/2$ and isolines of entropy variation (to highlight the tip leakage vortex) colored by ZDES mode.

## 7. Tip Leakage Flow

The contours of $\lambda$ sensor near the casing are plotted in Figure 10 for the three ZDES computations. As expected, the flow far from walls is treated by LES part of ZDES approach. For the simulation ZDES-SA, the RANS region is very small from the first axial plane, as it is limited to the casing boundary layer and the tip region. The tip leakage vortex is treated by LES from its appearance, once the flow exits the tip gap. The coherent aspect of the tip leakage vortex in the first plane is due to the necessary time for eddy viscosity destruction. Thus, the protection function $f_d$ works well for ZDES-SA. For both ZDES-SST and ZDES-BSL, the tip leakage vortex belongs also to the LES region (see plane 1). Nonetheless, RANS regions are more spread, beyond the tip flow and the casing boundary layer, especially in the vicinity of the pressure side. This is due to (i) the protection function $1 - F_F$ which is more conservative as the boundary layer thickness is overestimated and (ii) the behavior of $L_{\text{RANS}}$ depending on advected turbulent kinetic energy $k$ and specific turbulent dissipation $\omega$ while for ZDES-SA this length scale depends only on wall distance. Whatever the discrepancies between ZDES approaches, they behave well as boundary layers are well protected, the flow within the tip (from the third plane) and tip leakage vortex belong to LES region. Nevertheless, for the ZDES-BSL, the flow in the tip region is treated in LES only from the fourth plane because of the higher level of eddy viscosity due to the absence of SST correction and its impact on the RANS length scale. The comparison between the two blades highlights qualitatively the influence of external perturbations (IGV wakes here) on the protection function and on RANS and LES length scales. The approach ZDES-SA is clearly less sensitive than ZDES-SST and ZDES-BSL to these perturbations.

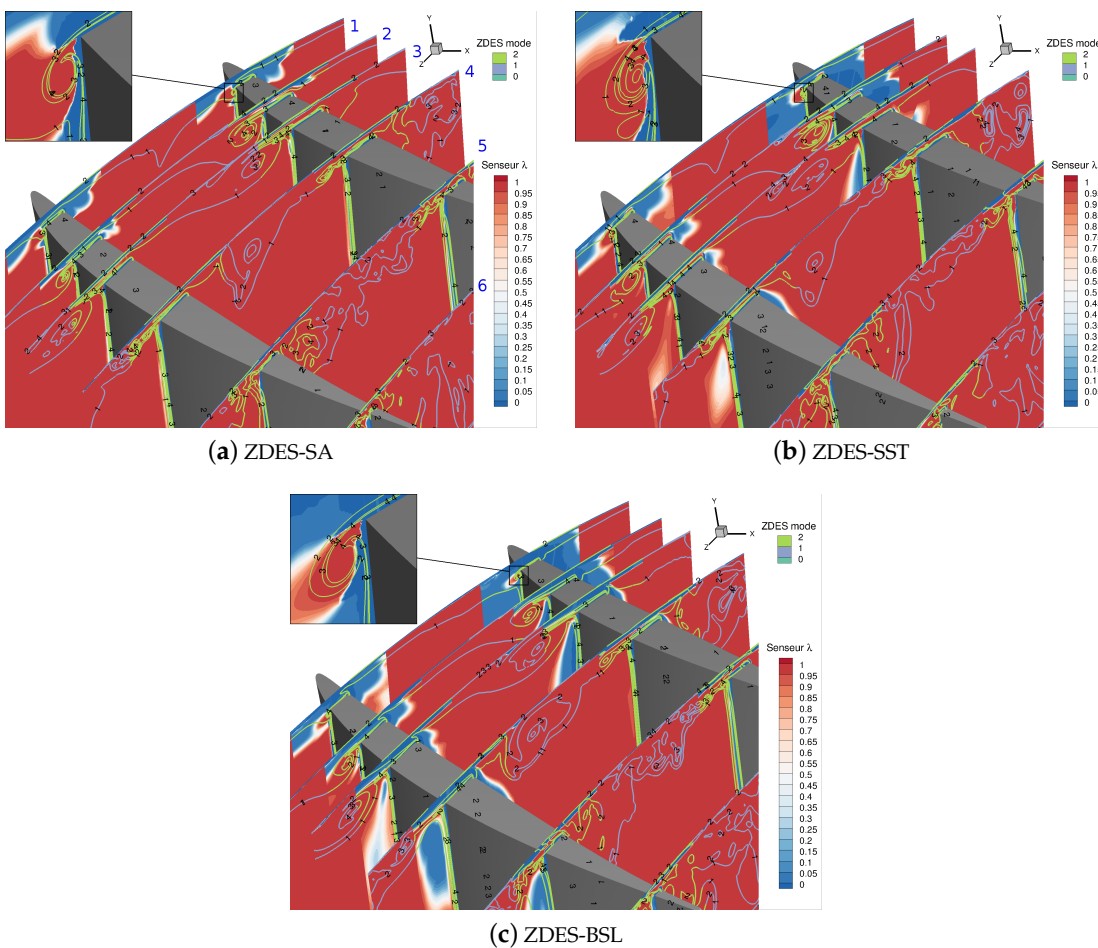

**Figure 10.** Contours of $\lambda$ sensor in different axial planes near the rotor blade tip at the instant $t = T/2$ and isolines of entropy variation (to highlight the tip leakage vortex) colored by ZDES mode.

Figure 11 depicts the isosurface of Q criterion colored by normalized helicity and entropy variation field in 31% x/c plane, for the five unsteady simulations, at the instant $t = nT$ (multiple of IGV passing period). Due to vortex dissipation, especially across the weak shock at 31% x/c, all unsteady RANS simulations are unable to capture any interaction between the tip leakage vortex and the tip flow of the adjacent blade. There is no double leakage flow phenomenon. On the contrary, all ZDES capture the double leakage flow even if the amplitude depends noticeably on the underlying RANS turbulence model. With SA and SST models, ZDES predicts a vortex breakdown in the vicinity of the adjacent blade, leading to numerous secondary vortices interacting with tip flow. Some of them are convected upstream to interact with the tip leakage vortex of the adjacent blade. For the BSL model, as the eddy viscosity is higher, the dissipation is increased, leading to a vortex breakdown located more upstream and secondary vortices are partially dissipated. Fewer secondary vortices interact with the tip leakage vortex. This analysis shows that the underlying turbulence model has a small impact on tip leakage flow by comparison to the switch from fully unsteady RANS to ZDES approaches.

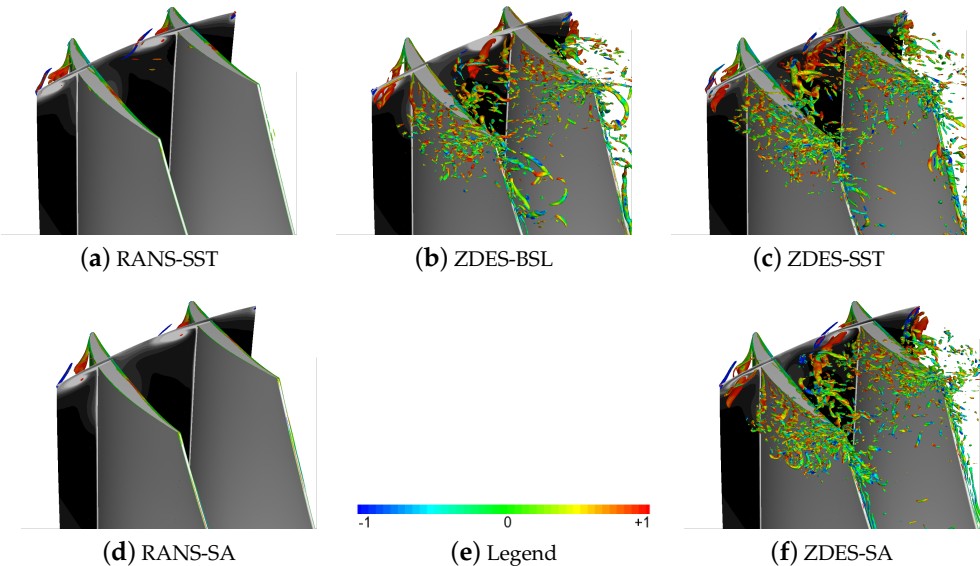

**Figure 11.** Isosurface of Q criterion colored by normalized helicity and entropy variation field in 31% x/c plane (grey color map). From top to bottom, from left to right: RANS-SST, RANS-SA, ZDES-BSL, ZDES-SST, and ZDES-SA.

## 8. Hub Corner Flow

Before analyzing the flow over the rotor blade and close to hub corner, the verification of LES behavior in ZDES computations is first discussed. The contours of the $\lambda$ sensor close to the rotor blade junction with the hub are depicted in Figure 12, at the instant $t = T/2$ and for the three ZDES simulations. Isolines of entropy variation colored by ZDES mode are also plotted to highlight the corner flows and separations. Whatever the formulation, the attached boundary layer over the hub or the rotor blades seems to be well protected for all ZDES (see planes 1 to 4). As shown by entropy variation isolines, the boundary layer separation at the hub corner provokes the switch to LES behavior of ZDES as wanted (planes 5 to 7). Nonetheless, for the BSL underlying turbulence model, the RANS region is more spread and the separation is treated with RANS equations (planes 5 and 6). This can be due to the absence of SST correction and higher eddy viscosity and to a too thin separation (recall that mode 2 is well suited for massive separation i.e., with a height significantly higher than the boundary layer thickness). When the separated region is sufficiently thick, the RANS behavior is replaced by the LES one (plane 7). Over the pressure side, as the protection function is more conservative, RANS regions are more widespread with SST and BSL turbulence models. As the spatial periodicity of IGV is twice the periodicity of the first rotor, the effect of IGV wakes and vortices is not the same for both rotor blades. The comparison between the two blades shows that the ZDES-SA is quite insensitive to external perturbations while both SST and BSL are significantly impacted by these perturbations.

As ZDES approach behaves as expected, the impact of turbulence modeling approach and underlying turbulence RANS model is discussed. The skin-friction lines and the skin-friction magnitude are shown in Figure 13 for the five unsteady simulations at the instant $t = \frac{T}{2}$. Although the instantaneous field is not representative of the time-averaged one (e.g., the instantaneous flow is not always separated or the time-averaged separated flow is less spread), and this gives a qualitative overview of separation footprint over walls. Unlike the tip flow, the underlying turbulence model influences significantly the flow near the rotor–hub junction where a hub corner separation occurs. This influence is observed whatever the used approach: RANS or ZDES. For unsteady RANS simulations, the most large separation is predicted by the SST model. The switch to ZDES approach has a strong influence on the topology of skin-friction lines and the skin-friction magnitude field. With the SA underlying turbulence model, the separation predicted by ZDES is much larger than the RANS

one. This statement is also valid for the SST model. The comparison between all ZDES predictions shows that the separation predicted with SST and BSL model spreads up to the blade tip. This leads to a thickening of the boundary layer above the first third of span as the skin-friction lines are mainly radially oriented. The most important separation is obtained with the BSL model, especially as the topology is composed of a focus which is the footprint of a tornado-like vortex which is representative of a larger separation [37].

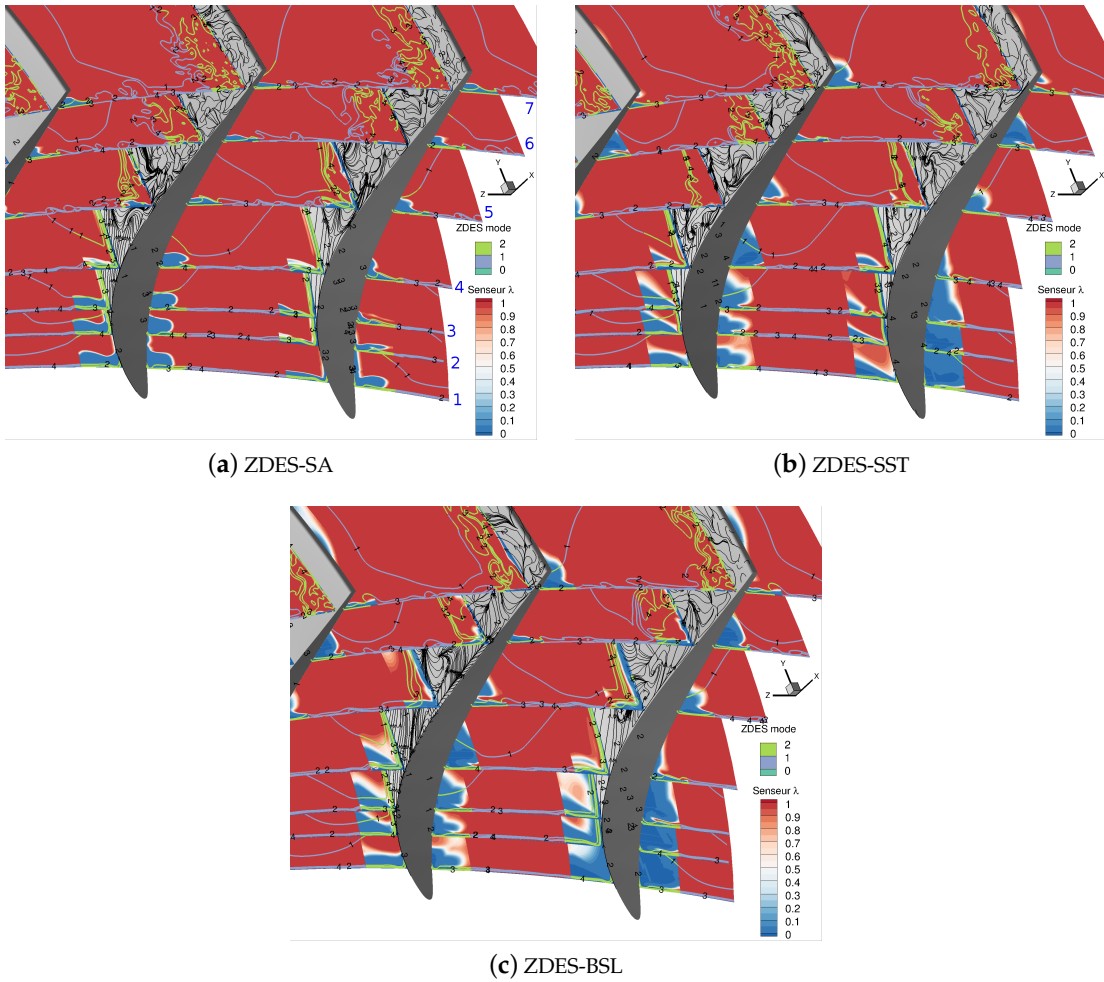

**Figure 12.** Contours of $\lambda$ sensor in different axial planes near the rotor blade junction with the hub at the instant $t = T/2$, isolines of entropy variation (to highlight boundary layers and separations expansions) colored by ZDES mode, and skin-friction lines over the rotor blades.

At the same instant of Figure 13, Figure 14 depicts the skin-friction lines topology and the skin-friction magnitude over the hub walls. The separation predicted by unsteady RANS simulations don't spread noticeably over the hub. Downstream the blade, the separated flow can be confused with the blade wake. With ZDES approach, the flow topology near the hub is much more complex due to the footprint of numerous coherent structures. The separation spreads significantly in the azimuthal direction. Contrary to the topology over rotor blade, the largest footprint of the separation over the hub is predicted by ZDES-SA, followed closely by ZDES-SST and far behind by ZDES-BSL. Taken with the previously commented separation footprint over the blade, this means that, for ZDES-SA, the separation is much more confined close to the junction while the ZDES-SST and ZDES-BSL predict a separation that is much more radially spread.

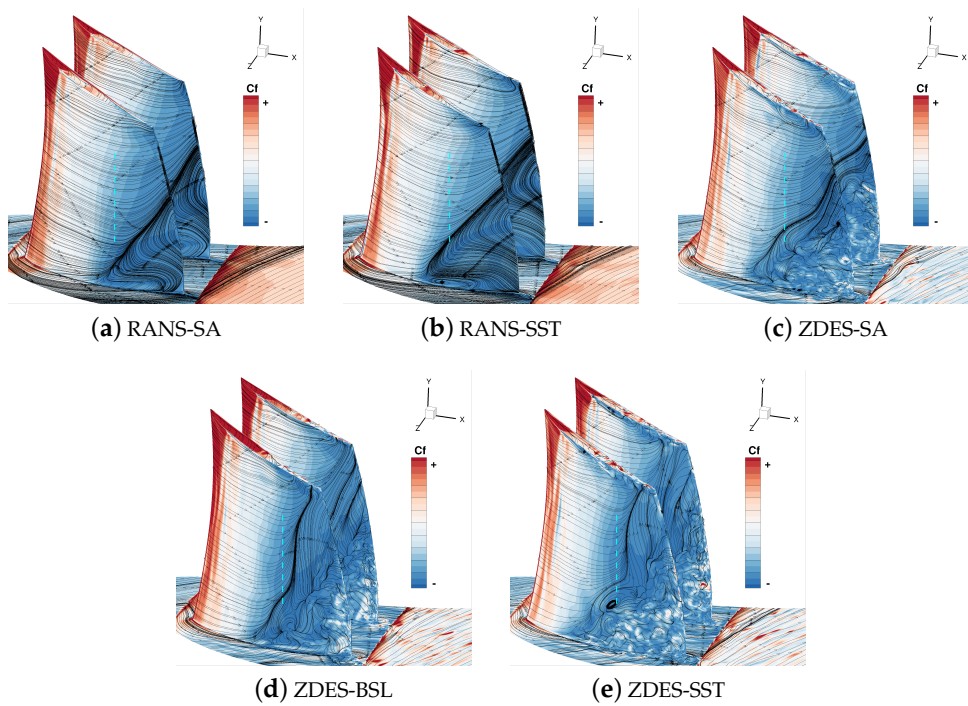

**Figure 13.** Skin-friction magnitude and skin-friction lines over hub and rotor blades at $t = T/2$.

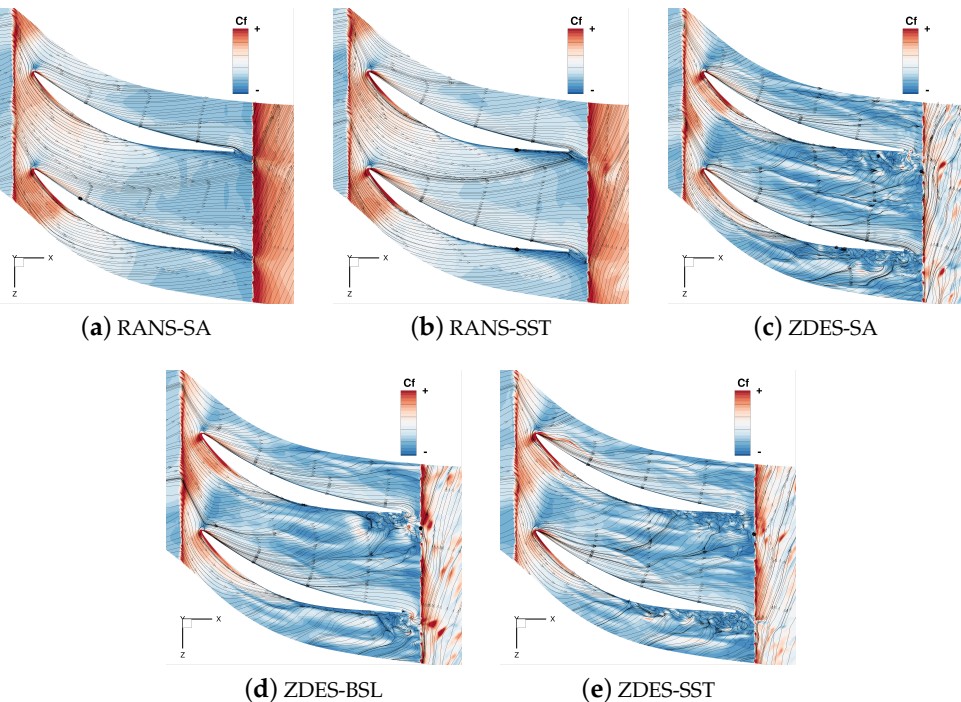

**Figure 14.** Skin-friction magnitude and skin-friction lines over the hub at $t = T/2$.

As all walls are adiabatic, the entropy variation is a good sensor of losses within a compressor. The entropy variation field in the axial plane located at 91% x/c, with $c$ the tip axial chord, is depicted in Figure 15 at the instant $t = 3T/4$. This field allows the visualization of the tip flow, the hub corner flow, and IGV vortices observable near the left rotor blade. Near the tip, the dissipation of tip leakage

vortex by RANS approach and the breakdown by the ZDES one are clearly observed. As shown previously, the separation is much more radially spread for SST and BSL models. The boundary layer is thicker with the $k - \omega$ turbulence model. Near the hub, the ZDES-SA separation is more massive than the RANS one. This is also observed with the SST turbulence model. The entropy field shows the confinement of separation for ZDES-SA by comparison to ZDES-SST and ZDES-BSL. Near the hub, the separation predicted by ZDES-SST and ZDES-BSL is less azimuthally and more radially spread than the ZDES-SA one. The comparison between ZDES-SST and ZDES-BSL shows that the higher level of eddy viscosity in the attached boundary layer with the BSL model is responsible for a RANS to LES switch more downstream which delays the chaotic degeneration of the corner separation's shear layer and the turbulent fluctuations' appearance in the separated flow.

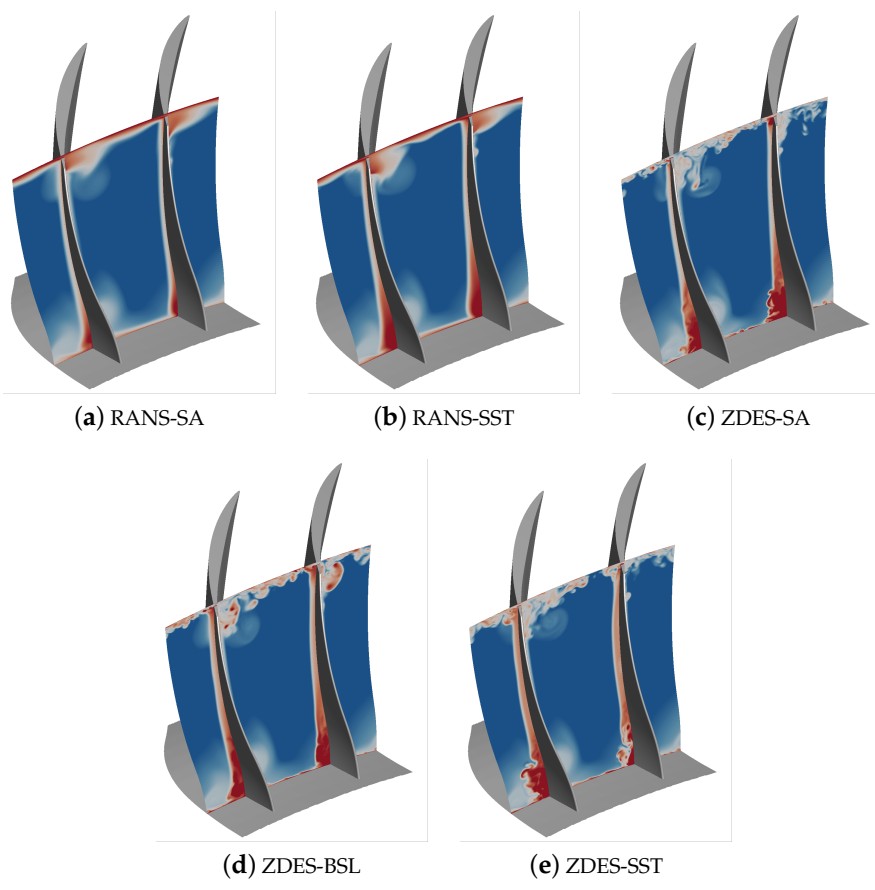

**Figure 15.** Entropy variation field in plane located at 91% x/c at $t = 3T/4$.

The previous analysis is quite qualitative as only one instant is investigated and this is not representative of time-averaged flow field. The time-averaged field of entropy variation in several axial planes, 21.7%, 31%, 44.8%, 67.8%, 91%, and 109% x/c, is shown in Figure 16. It should be noticed that, due to the time-averaged, the influence of external perturbations is not observable. As in instantaneous data, the tip leakage vortex predicted by RANS approach, for both SA and SST turbulence models, is clearly dissipated from the plane located at 31% x/c. Near the tip, losses are very similar between these two models (levels and high entropy area). Near the hub, losses are significantly higher for the SST model due to a larger separated region coming from the blade–hub junction. Moreover, the boundary layer is thicker for this model. As simulations are based on RANS approach, similar conclusions to instantaneous analysis were expected. For ZDES computations, despite the time average, the tip leakage vortex is clearly visible, especially its trajectory. This trajectory depends more on the approach (RANS vs. ZDES) than the turbulence model (SA vs. SST). With ZDES,

the vortex is more oriented towards the adjacent blade which promotes the double leakage flow phenomenon. The absence of this phenomenon in RANS simulations allows the visualization of the tip leakage vortex footprint in the downstream plane (foreground plane) while, in ZDES, the vortex can not be distinguished from the rotor wake. Near the hub, the analysis performed for only one instant is still valid. With the SA model, the ZDES amplifies the hub corner separation with respect to RANS approach. On the contrary, with the SST model, the most massive separation is obtained with the RANS approach. Thus, the underlying turbulence model has a strong influence on the junction flow and on corner separation. The comparison of all simulations highlights a hierarchy between them in terms of apparent size and magnitude of loss pattern due to hub corner separation: RANS-SA < ZDES-BSL < ZDES-SST < ZDES-SA < RANS-SST.

Another concern is the influence of external perturbations due to relative location with respect to secondary flows coming from IGV. The comparison between the two blades shows that the ZDES results are sensitive to these perturbations contrary to the unsteady RANS approach. This is clearly observed in Figures 13 and 14. The instantaneous field of entropy variation shows that: (i) IGV tip vortex influences the tip leakage vortex, especially its convection and breakdown, (ii) with ZDES approach, hub corner separation is more sensitive to the IGV vortex as the interaction between this vortex and the junction flow is responsible for a more massive separation. This last statement concerns both SST and SA underlying turbulence models. Although this must be analyzed in stage configurations to be confirmed, the flow around the rotor is more impacted by rotor–stator interactions with ZDES approach.

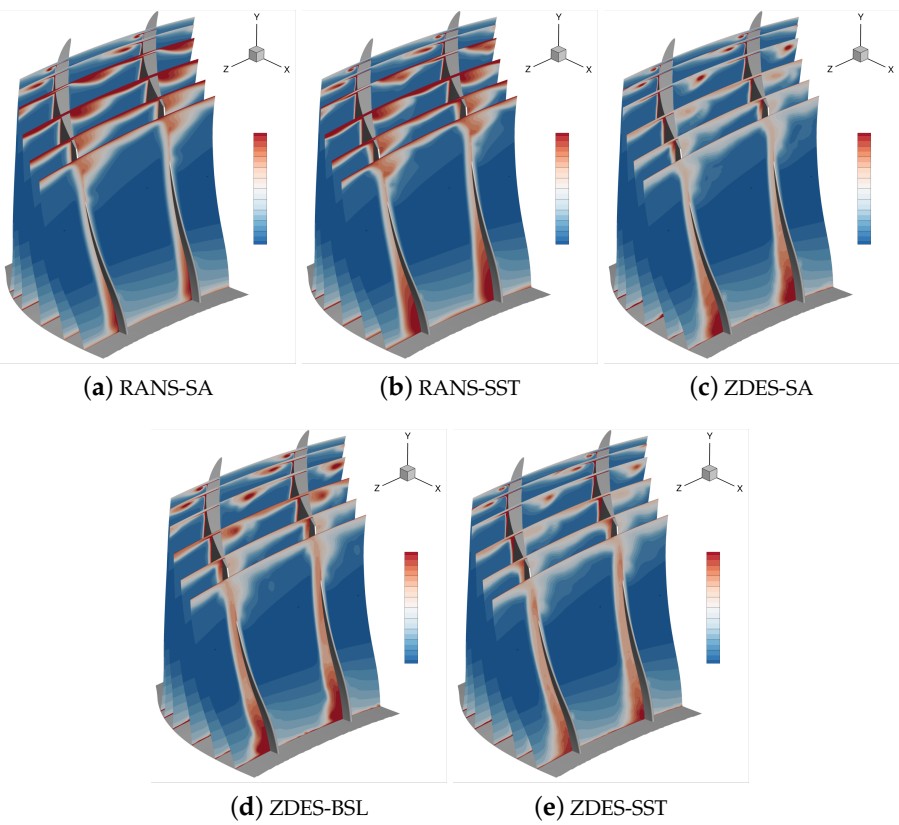

**Figure 16.** Time averaged field of entropy variation in different axial planes (21.7%, 31%, 44.8%, 67.8%, 91%, and 109% x/c, with c the tip axial chord).

## 9. Conclusions

The present study focuses on the impact of the underlying RANS turbulence model in the Zonal Detached Eddy Simulation (ZDES) method when used for prediction of secondary flows around a rotor blade of HP compressor. This is carried out in light of the issues commonly investigated for hybrid RANS/LES methods: detection and protection of attached boundary layer (ⓐ in the introduction), emergence and growth of resolved turbulent fluctuations (ⓑ) and accurate prediction of separation front due to progressive adverse pressure gradient (ⓒ).

All simulations are validated with experimental data and are in good agreement with measurements. The main discrepancies between the experimental and the numerical configurations come from the absence of axial gaps, the underestimation of axial momentum close to the hub and the underestimation of stagnation pressure close to both endwalls in the inlet plane. It must be pointed out that the lack of measurements in region of interest (here the hub/blade junction) is a limiting difficulty for discriminating the considered approaches and models. It prevents a thorough assessment in respect to issue ⓒ. The two ZDES based on the $k - \omega$ model of Menter [25] improve the agreement in terms of radial gradient of stagnation pressure, especially the negative gradient far from endwalls.

At mid-span, the boundary layers seem to be well protected. There is no evidence that separations are influenced by any model-stress depletion, grid induced separation or by large over protection of boundary layer. The wake is solved by LES from the trailing edge. Near the tip, the impact of the approach is clearly shown on the tip leakage vortex, whatever the underlying turbulence model which has a small effect. The tip leakage vortex is treated as an LES region allowing emergence of resolved turbulent content. The boundary layers over tip, pressure, and suction sides are well protected. Near the hub, the underlying turbulence model has a strong influence on the junction flow development, especially the corner separation. This is also impacted by the SST correction and subsequently by the eddy viscosity level as shown by the comparison between ZDES-BSL and ZDES-SST. Finally, whatever the underlying turbulence model, the ZDES behaves well with respect to issues ⓐ and ⓑ as the boundary layers appear effectively shielded and the RANS-to-LES switch is close downstream of trailing edges and separation fronts leading to a quick LES treatment of wakes and shear layers. This allows an early apparition and growth of resolved turbulent content.

The comparison between RANS and ZDES approaches show that only the ZDES is able to accurately predict the tip leakage vortex in the whole passage while RANS approach dissipates it. This error is noticeable in the time-average field as RANS and ZDES predict different flows. Only the ZDES approach is well suited for the tip leakage flow prediction. The impact of underlying turbulence model is smaller near the tip. Near the blade–hub junction, the underlying turbulence model has a strong influence which modifies the corner separation prediction. In the present study, the ZDES method with SA underlying model predicts a larger separation than ZDES with SST or BSL underlying models considering the apparent size and magnitude of the corner separation loss pattern. Following this latter qualitative criterion, a hierarchy between all simulations appears: RANS-SA < ZDES-BSL < ZDES-SST < ZDES-SA < RANS-SST. However, the corner separation topologies are also different: with ZDES-SST and ZDES-BSL its greater extension along blade height leads to a radial redistribution of losses and so to total pressure radial gradient closer of experimental measurements. In summary, the RANS underlying model has no significant influence on the tip flow, as the tip gap can be viewed as a geometric singularity. However, this model influences the growth of attached boundary layers and the corner flow. In other words, prediction of corner separation depends significantly on the underlying turbulence model.

Subsequent to this investigation, as the underlying turbulence model impacts corner flow prediction, the most important open question concerns this flow, especially corner separations. Thus, hybrid RANS/LES methods must be more deeply studied and validated regarding corner flows. For that, it is necessary to simulate configurations for which detailed measurements are available (the test rig of Zambonini et al. [38] for example). Moreover, several items can be investigated: (i) underlying turbulence model, with Differential Reynolds Stress Model for example, (ii) influence

of resolved turbulent dynamics in incoming boundary layer, with the mode 3 of ZDES, and (iii) the impact of leakage flow due to axial gap which influences the flow near the hub, especially the flow angles and stagnation pressure.

**Author Contributions:** Funding acquisition, J.M.; Investigation, J.M. and C.U.; Methodology, J.M.; Project administration, J.M.; Validation, J.M. and C.U.; Visualization, J.M. and C.U.; Writing—original draft, J.M.; Writing—review and editing, J.M. and C.U. All authors have read and agreed to the published version of the manuscript.

**Funding:** This work was granted access to the HPC resources of CINES and GENCI (allocation A1-A0012A10078).

**Acknowledgments:** All simulations have been performed in the framework of the *elsA* three-party agreement between AIRBUS, SAFRAN, and ONERA, which are co-owners of this software. The authors wish to thank William Riéra, who carried out the presented numerical test bench and RANS-SA/ZDES-SA simulations. The authors thank Georges Gerolymos, Lionel Castillon, and Sébastien Deck for useful discussions during this study.

**Conflicts of Interest:** The authors declare no conflict of interest. The funders had no role in the design of the study; in the collection, analyses, or interpretation of data; in the writing of the manuscript, or in the decision to publish the results.

## Abbreviations

The following abbreviations are used in this manuscript:

| | |
|---|---|
| BSL | BaSeLine Menter turbulence model |
| HRLES | Hybrid RANS/LES |
| IGV | Inlet Guide Vane |
| LDA | Laser Doppler Anemometry |
| LES | Large Eddy Simulation |
| R1 | First rotor of CREATE compressor |
| RANS | Steady/unsteady Reynolds-Averaged Navier–Stokes |
| SA | Spalart–Allmaras turbulence model |
| SST | Menter turbulence model with SST correction |
| ZDES | Zonal Detached Eddy Simulation |

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
