# Peer review of "Impact of Underlying RANS Turbulence Models in Zonal Detached Eddy Simulation: Application to a Compressor Rotor"

_ijtpp, doi:10.3390/ijtpp5030022_

Round 1

Reviewer 1 Report

(1) please add the discussion of inlet flow profile and dicrepancies with experiments at 25A into the main body of the paper. This is the main cause of overestimation of corner flows.

(2) please summarise this cause of discrepancy in the conclusion section. 

Author Response

The authors thank the reviewer for his relevant comments and recommendations. The answers are given below.

(1) Inlet maps and flow profiles are added in the part 3, showing that some discrepancies are already observed at inlet plane, close to endwalls. These information are used in discussion in part 5.

(2) The conclusion section is consequently modified.

Reviewer 2 Report

Introduction

The introduction needs few improvements. As general comment none of the statements is supported by references or other type of evidence. It is a very generic discussion: readers not familiar with the problem will struggle to understand, while proficient readers will not find it useful at all. It is also lacking some structure in order to show more clearly (1) what the problem to be looked at is, (2) what the current status is in research or industry and (3) therefore what and how the paper aims to contribute to in this context.

Reference to other works on similar subject is almost absent.

Some simple examples of parts from the introduction that need to be reconsidered:

line 16: "The corner separation is another main secondary flow." This statement really does not add much.

line 18: "As the flow is unsteady and vortical..... On the contrary LES is a good candidate". Unsteady and vortical does not mean much (especially vertical: all viscous flows have vorticity!). Why LES should be better is not substantiated or supported.

Line 28-30: all sentence is very confusing and almost cryptic

Section 2: Experimental Facility

As for introduction is there any reference or published material about experimental set-up, facility and measuring instruments? That would be worth adding.

Section 3: Numerical test bench

Line 58-61: a bit confusing and rephrasing is needed. Apart from local effects due to the hub-flow cavity interaction, the leakage flows are responsible for stage matching and operating point distribution in a multi-stage compressor. In the present comparison the single rotor operating condition is anyway matched to exp data upstream and downstream and this should be mentioned as well to reassure that the rotor is as much as possible on target with rig data. (i.e. the discussion that follows about BCs).

A mention about inflow turbulence and the absence of fluctuations at inflow would be useful. I believe there are no turbulent flow fluctuations or any sort of synthetic turbulence entering the domain. I assume the precursor IGV-R1 simulation is a URANS.

Section 4: Zonal Detached Eddy Simulation

Line 97-100: the whole sentence is not clear while some references to terms such as “geometry” or “massive” refers to a classification probably hinted in Figure 2 but not explained or properly referenced.

Line 108-114: As above this part is not clear. I acknowledge that the variants of hybrid methods in literature are per-se very confusing, but this section could be a nice opportunity to make it a bit clear at least in relation to the variants here explored. The distinction between scales (Ldes, Lles and Lrans) is unclear in this context and table 1, although being a good idea to use a table actually does not make it clear at all. I would suggest reviewing this part trying to make it more systematic and clear. At the end of the section the reader should have clearly understood what LES model will be used and where, eventually, detailed info can be found. Using references are ok.

Section Rotor Performance.

Line 148-153: this sentence is not clear starting with the assert about radial distribution and its relation to small and large compressions at casing and hub. Not sure what it means and therefore the following sentence about global positive gradient observed in Fig 3b is not making much sense either.

Line 159: the difference to exp at hub is addressed between different models. It would be work recalling the assumption that there is not hub cavity flow in the simulations and therefore comment about the fact that this has an impact on the difference between simulations and exp.

Hub Corner flow

Generally is ok, but a bit of review of the text to easy the reading would be very beneficial. Try to simplify concepts in each sentence and avoid too much information at the same time.

Line 341: the conclusion is not clear. Can you explain better? I am sure I am wrong, but it does look like contradicting the final statement in the conclusion: RANS-SA has the smaller separation, but in conclusion is said to be the largest (line 367).

Conclusions.

Generally ok, but it would be very useful to make a link with the objectives in the Introduction and summarize more clearly the conclusion, limitations or recommendations obtained with this work.

That will also provide a nice slide into the future and still open question that is then considered “Subsequent to this investigation…” and proposed in the final section.

Author Response

The authors thank the reviewer for his relevant comments and recommendations. The answers are given below. All removed text is in red and new text in blue.

Introduction

The introduction has been reviewed, reorganized and some parts re-written. Numerous references are added and current status in literature is more clearly shown. 

Lines 16 and 18 are modified following the reviewer remarks.

Section 2: Experimental Facility

The experimental set-up and measurements are more detailed. References are also added.

Section 3: Numerical test bench

The reviewer is right concerning the stage matching in a multi-stage compressor which is influenced by leakage (due to tip gap or axial gap at the hub) and corner flows. In the present case, the hub cavity is not taken into account, which can explain locally some discrepancies near the hub. However, inlet boundary conditions are set so that the single rotor operating point matches as much as possible the experimental one. It should be noticed that the static pressure used for the radial equilibrium law is defined so that the radial profile of axial momentum at the inlet matches the experimental one near the casing.

The reviewer is right. There is no turbulent fluctuation at the inlet so that boundary conditions are identical between RANS and ZDES computations. The “RANS” term is mentioned for the precursor simulation.

Section 4: Zonal Detached Eddy Simulation

The definition of massive separation is given. The “geometry” term is replaced by “geometric singularity” (for example, backward facing step). Definitions of ZDES modes are shortly described. The reference “Deck & Laraufie, 2013” is also added to illustrate differences between modes.

The SGS model is mentioned. The RANS-to-LES switch and length scales are detailed.

Section Rotor Performance.

As these sentences do not add significant information in the present article, they are removed.

The absence of hub cavity is recalled at the end of first paragraph and is mentioned in discussion about discrepancies between CFD and experimental data.

Hub Corner flow

In order to better introduce some concepts or to better discuss them, some sentences are reviewed. Moreover, this part is composed of three subsections: verification of shielding function, discussion on corner separation and influence of wakes and vortices coming from IGV.

In conclusion, the largest separation is predicted by ZDES-SA. There is no contradiction with the final statement of “hub corner flow” part. The hierarchy with RANS approach is now added in conclusion.

Conclusions.

With respect to the three issues encountered with HRLES methods mentioned in Introduction, whatever the RANS underlying model, the ZDES behaves well as the boundary layer is well-shielded and the RANS-to-LES switch is close to trailing edge leading to a LES treatment of wakes.

In summary, the RANS underlying model has no significant influence on the tip flow, as the tip gap can be viewed as a geometric singularity. However, this model influences the growth of attached boundary layer and the corner flow. In other words, prediction of corner separation depends on the RANS underlying model.

The major open question is discussed in conclusion and further works are mentioned.